# Implantation of a Leadless Pacemaker after Incomplete Transvenous Lead Extraction in a 90-Year-Old Pacemaker-Dependent Patient

**DOI:** 10.3390/ijerph19106313

**Published:** 2022-05-23

**Authors:** Gerald Drożdż, Bruno Hrymniak, Bartosz Biel, Przemysław Skoczyński, Wiktoria Drożdż, Dorota Zyśko, Waldemar Banasiak, Dariusz Jagielski

**Affiliations:** 1Department of General Internal Medicine, Kantonsspital Graubünden, 7000 Chur, Switzerland; wiktoria.k.g@gmail.com; 2Centre for Heart Diseases, Department of Cardiology, 4th Military Hospital, 50-981 Wroclaw, Poland; bruno.hrymniak@gmail.com (B.H.); bartosz.biel@gmail.com (B.B.); przeskocz@tlen.pl (P.S.); dzysko@wp.pl (D.Z.); banasiak@4wsk.pl (W.B.); dariusz.jagielski@gmail.com (D.J.); 3Department of Emergency Medicine, Wrocław Medical University, 50-367 Wroclaw, Poland

**Keywords:** transluminal lead extraction, leadless pacemaker, MICRA VR

## Abstract

Transluminal lead extraction (TLE) is a well-established procedure for the removal of damaged or infected pacing systems. Despite its high efficacy, the procedure is associated with significant risks, some of which may contribute to severe life-threatening complications. Herein, we present the case of a 90-year-old female who was 100% pacemaker-dependent (PM-dependent) and had ventricular lead fragmentation after the TLE procedure. In this elderly patient, after taking into account the whole clinical context—age, frailty syndrome, infection, and high peri- and postprocedural risks—we decided on MICRA VR implantation as well as leaving the remains of the ventricular lead in the right heart chambers. A Leadless pacemaker (LP) is an excellent alternative to PM-dependent individuals, in whom implantation of permanent transvenous PM is precluded due to multiple infectious and non-infectious issues.

## 1. Introduction

In recent years, due to rapid technological and medical progress, the number of indications for cardiovascular-implantable electronic devices (CIEDs) in treating bradycardia and tachycardia events has increased [1]. However, the implantation of CIEDs carries several non-negligible risks [2]. Among them, CIED infections are the leading complication after implantation [3]. Once diagnosed, blood culture sampling, initiation of intravenous antibiotic therapy, and subsequent CIED removal are inevitable [4]. TLE is a well-established procedure of damaged or infected pacing systems, which despite its remarkable efficacy, poses a risk of several procedure-related complications [5]. We present a case of a 90-year-old pacemaker-dependent female with ventricular lead fragmentation after the TLE procedure.

## 2. Case Presentation

The 90-year-old female that was 100% pacemaker (PM)-dependent with a clinically symptomatic complete heart block was hospitalized for a TLE procedure of dual-chamber PM due to the pocket infection and positive blood culture for methicillin-susceptible Staphylococcus Aureus (MSSA). The extraction was performed with a lead locking device introducer and Evolution 11 mechanical system and resulted in the atrial lead being fully extracted. A 15 cm-long fragment of ventricular lead was left behind between the apex and the border of the right atrium and vena cava superior. The microbiological sampling from the extracted lead’s endings grew fluconazole-sensitive Candida albicans. After the procedure, the patient was transferred to a remote hospital to complete a 3-week antimicrobial treatment with rifampin, fluconazole and vancomycin. During this period, the patient was placed on temporary transvenous pacing (Figure 1).

After completing antibiotic therapy, the patient was re-admitted for further management. The pre-procedural transthoracic echocardiogram (TTE) revealed preserved ejection fraction, moderate tricuspid regurgitation, and absence of pericardial effusion. A well-demarcated temporary intravenous pacing lead and the fragment of the ventricular lead’s sheath were disclosed in the right heart chambers without endocarditis suspicious vegetations. After taking into account the whole clinical context—age, multiple comorbidities (Type 2 diabetes mellitus, chronic kidney disease, venous thrombosis in left subclavian, axillary and brachial veins), frailty syndrome (moderately frail patient according to the clinical frailty scale), infection, and high peri- and postprocedural risks, the patient was qualified for LP implantation in the wake of risk–benefit evaluation. A decision of leaving the remains of the ventricular lead in the right heart chambers has been made. After adequate preparation, implantation of MICRA VR from the right femoral vein approach in local anesthesia was performed (Figure 2). The initial parameters were adequate, with a threshold of 1.0 V/0.2 ms, an R-wave of 9.6 mV, and an impedance of 610 Ohm. The radiation exposure dose was determined as 91.4 mGy, and the captured fluoroscopy time was 34 min. Subsequently, a temporary intravenous pacing lead was removed. No peri- or post-procedural complications were recorded.

## 3. Discussion

In 2009, Wilkoff et al. identified factors predicting the ease of the CIED extraction, stating that single-lead CIEDs of short implantation duration and devices with “young” non-ICD leads are expected to be more easily removed [5]. The primary PM implantation in 2004, passive fixation electrodes, and exchange in 2012 made the procedure challenging even for an experienced center. Despite careful management and an experienced team, lead fragmentation as well as other possible complications must have been taken into account.

The presented scenario implies several problems. First of all, the fragment of the lead remaining in the right heart chambers may cause hemodynamic consequences resulting in significant tricuspid regurgitation. Secondly, the left lead`s fragment should be considered a foreign body that imposes an increased risk of further infectious complications, with endocarditis being the most severe. On the contrary, another TLE procedure attempt could result in further lead fragmentation and subsequent pulmonary embolism. Finally, the open surgical approach under general anesthesia could be considered the ultima ratio. However, the patient`s advanced age and multiple comorbidities pose a high perioperative mortality risk.

In post-procedural follow-up, TTE excluded all possible hemodynamically significant problems. We have attributed this to the fact that the ventricular lead’s fragment consisted solely of the lead’s sheath, made of flexible silicone fabric, and thus unlikely to cause hemodynamic consequences. The common practice after infected CIED extraction is contralateral CIED reimplantation. Unfortunately, this carries an elevated risk of new infectious and thromboembolic complications [3]. In this elderly patient, after taking into account the whole clinical context—age, frailty syndrome, infection, and high peri-, and postprocedural risks—we decided on MICRA VR implantation. After a literature search, to the best of our knowledge, there is no other case of MICRA VR implantation and leaving the fragment of initial ventricular lead in the right heart chambers.

LP is an excellent alternative for PM-dependent individuals, in whom implantation of permanent transvenous PM is precluded due to multiple infectious and non-infectious issues [6]. The recent studies comparing LPs with lead PM systems revealed benefits from leadless devices, excluding the risk of lead or pocket-related infections and reducing the thromboembolic risk [7,8].

This case emphasizes the recent trend of patient-centered medicine, in which an individual’s health needs and desired outcomes have become the driving force behind all the medical decisions.

To date, there is a lack of reports describing the usage of a LP shortly after incomplete TLE due to local pocket infection or cardiac device-related infective endocarditis in a geriatric population with frailty syndrome. Usage of the LP in a pacemaker-dependent 90-year-old woman burdened with frailty syndrome, concomitant local pocket infection and part of an electrode stuck in a vein system/cardiac cavities during extraction became a solution for a very difficult clinical situation. In the presented case, the risk of potentially lethal complications (infectious or thromboembolic due to eventual cardiac surgery intervention as a consequence of bleeding or vascular or cardiac cavity damage) highly outweighed the possible benefits of a new trial of complete lead removal procedure.

## 4. Conclusions

The decision to leave a small fragment of the ventricular electrode of the infected CIED system, 3-week targeted antibiotic therapy, and LP implantation in 3-month follow-up proved to be an effective method of treating an aged patient in whom the next procedure of removing the electrode fragment could be associated with an unacceptably high risk of complications. Due to a lack of clinical and echocardiographic features of invasive candidemia, including no signs of infectious endocarditis, as well as negative control blood samples, the cultivation of Candida albicans from the electrode sampling was interpreted as contamination. Following that, there was no indication for further life-long suppression antimicrobial treatment. In the case of younger patients with a lower risk of complications, it would be advisable to complete full removal of an infected system.

## 5. Take-Home Message

While performing TLE, a possible variety of complications, such as lead fragmentation, must be taken into account.In elderly PM-dependent patients, careful risk–benefit evaluation should precede a new trial of fragmented lead extraction.LP is an excellent alternative for PM-dependent individuals who manifest contraindications for permanent transvenous PM implantation.

## Figures and Tables

**Figure 1 ijerph-19-06313-f001:**
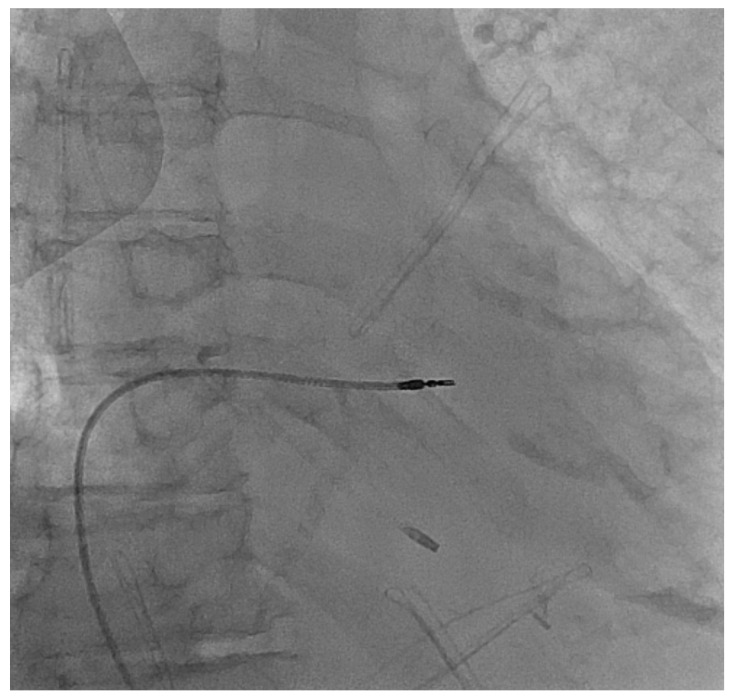
Chest X-ray in posteroanterior view with temporary pacing electrode introduced from right groin and remains of the ventricular electrode after incomplete transvenous lead extraction.

**Figure 2 ijerph-19-06313-f002:**
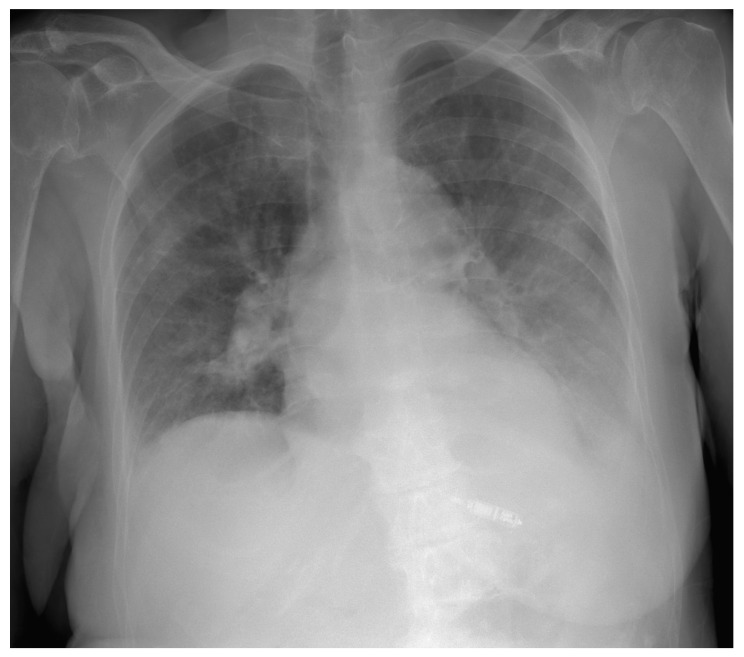
Chest X-ray in posteroanterior view with implanted leadless pacing system in the interventricular septum in the right ventricle.

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
