# Peer review of "Implantation of a Leadless Pacemaker after Incomplete Transvenous Lead Extraction in a 90-Year-Old Pacemaker-Dependent Patient"

_ijerph, 2022, doi:10.3390/ijerph19106313_

Round 1

Reviewer 1 Report

Line 14 - a 90-year old female that was 100% pacemaker dependent (PM-dependent) and had ventricular lead fragmentation after TLE procedure.

Line - 28  - Once diagnosed , blood culture sampling, initiation of intravenous antibiotic therapy, and subsequent CIED removal are inevitable [4].

 ---- consider citing “ European Consensus Document on Cardiac Implantable Electronic Device Infections” 2020

Line 35 – consider rephrasing as previously noted “ The 90-year old pacemaker (PM) dependent”

Line 51 – “The patient was qualified for LP implantation in the wake of risk-benefit evaluation.

With regards to the case presentation :

»»» Please mention in the case presentation the clinical context in this patient that led to the decision in question. What were the comorbidities and frailty scale ?  - age, frailty syndrome, infection, and high peri-, and postprocedural risks.

»»» Antibiotics used should also be mentioned.

»»» How was the patient managed regarding the Candida albicans isolation ? Was there evidence of invasive candidaemia ? And did she receive life-long suppression therapy with a triazole ?

Reviewer 2 Report

The authors present a case study for implantation of a leadless pacemaker after incomplete transvenous lead extraction in a 90-year-old pacemaker dependent patient. The paper is well prepared, and the case is present clearly.

However, novelty in the case study requires further explanation.

What makes this study distinguishing from other pacemaker removal cases?

Emphasis on the case is required in the Discussion section.

Round 2

Reviewer 2 Report

The authors respond to the reviewer's comments sufficiently. The submitted paper can be accepted as a publication.